# Adsorption of Diclofenac Sodium by Aged Degradable and Non-Degradable Microplastics: Environmental Effects, Adsorption Mechanisms

**DOI:** 10.3390/toxics11010024

**Published:** 2022-12-27

**Authors:** Siqi Liang, Kangkang Wang, Kefu Wang, Yuli Kou, Tao Wang, Changyan Guo, Wei Wang, Jide Wang

**Affiliations:** 1Key Laboratory of Oil and Gas Fine Chemicals, Ministry of Education & Xinjiang Uygur Autonomous Region, School of Chemical Engineering and Technology, Xinjiang University, Urumqi 830000, China; 2Department of Chemistry and Centre for Pharmacy, University of Bergen, 5007 Bergen, Norway

**Keywords:** ultraviolet radiation, diclofenac sodium, aged microplastics, adsorption mechanisms

## Abstract

Microplastics (MPs) are novel pollutants, which can carry toxic contaminants and are released in biota and accumulate. The adsorption behavior of MPs and aged MPs has attracted extensive attention. In this paper, the aging process of polystyrene (PS) and poly (butyleneadipate-co-terephthalate) (PBAT) plastics under ultraviolet (UV) irradiation at a high temperature and their adsorption properties for the contaminant diclofenac sodium (DCF) before and after aging was investigated. There are many factors affecting the adsorption capacity of MPs. In this experiment, three aspects of MPs, organic pollutants, and environmental factors are explored. The Freundlich model as well as the pseudosecondary kinetic model is more applicable to the process of DCF adsorption by MPs. The main effects of adsorption of organic pollutants by MPs are electrostatic interactions, hydrogen-halogen bonds, and hydrophobic interactions. The adsorption capacity of the UV-aged MPs on DCF is significantly enhanced, and the order of adsorption capacity is Q_(A-PBAT)_ (27.65 mg/g) > Q _(A-PS)_ (23.91 mg/g) > Q _(PBAT)_ (9.30 mg/g) > Q _(PS)_ (9.21 mg/g). The results show that more active sites are generated on the surface of MPs after aging, which can enhance their adsorption capacity for organic pollutants. This adsorption mechanism will increase their role as contaminant carriers in the aquatic food chain.

## 1. Introduction

Plastics are widely used by humans due to their low cost, lightweight, and strong durability; global plastics production is estimated to have reached 348 million tons in 2017 [1,2]. Plastics and their products are mostly disposable products, which can easily cause environmental pollution due to their persistence by discarding the used plastics indiscriminately. Plastic debris undergoes chemical, physical, and biological degradation to form tiny plastic fragments. When the particles are less than 5 mm in diameter, these are called microplastics (MPs) [3]. MPs are widely present in a variety of environments, and their potential impact on the environment is being extensively studied [4]. The fate of MPs in the environment has been previously investigated in the literature, such as spatial distribution [5], biotoxicity [6], and transport [7]. MPs are not only directly harmful, but when they adsorb pollutants, they transport them into the environment through migration and diffusion, causing more serious pollution [8]. More importantly, MPs’ adsorption of pollutants may affect the overall toxicity of the mixture due to synergistic or antagonistic effects [9], which can cause harm to human health through the food chain. Therefore, it is crucial to predict the impact of MPs on the environment by studying the interaction of MPs with pollutants.

The properties of MPs can be altered by sunlight irradiation. Ultraviolet (UV) irradiation alters the polymer structure by increasing the oxygen-containing functional groups, resulting in chain breaks [10]. While the chemistry of MPs present in the environment differs from the plastic models commonly used in laboratory studies, which are typically pristine particles with uniform shape and size, the same aging effect as in the environment can also be achieved by UV aging experiments in the laboratory. Therefore, attention has been paid to the properties and pollutant adsorption of MPs in a UV aging simulation environment. Vroom et al. showed that the aged polystyrene MPs are more easily ingested by zooplankton [11]. Zhang et al. confirmed the improved adsorption of organic triclosan by aged polyethylene [12]. In contrast, Huffer et al. showed a decreasing trend in the adsorption properties of aged polystyrene MPs for different organic compounds [13]. The above results indicate that UV aging has a significant effect on the performance of MPs, especially on the adsorption performance of pollutants. Although some studies have focused on this field, more research is needed to summarize the impact rules and thus, to solve the environmental problems caused by MPs.

The largest categories of total global non-fibrous plastics production are estimated to be polyethylene (PE) at 36% and polystyrene (PS) at 10% [14]. Both are the most common MPs in the aqueous environment [15]. In addition, biodegradable plastics such as poly (butylene adipate-co-terephthalate) (PBAT) are beginning to be widely used in agricultural mulch, cling film, and plastic bags. The annual production of biodegradable plastics is estimated to be about 2.1 million tons, which is expected to increase to 2.4 million tons per year by 2025 [16]. Therefore, it is important to investigate the effects of two common environmental MPs (PS and PBAT) before and after aging on the sorption of organic pollutants.

Personal care products (PPCPs) and pharmaceuticals are frequently detected in the natural waters [17]. Adsorption of PPCPs in soil/sediment is a major process affecting their mobility and ultimate fate in the environment [18]. Among them, diclofenac (DCF) is widely used in PPCPs, which is an anti-inflammatory compound for humans and animals. Appendix A describes the structure and properties of DCF. The global annual consumption of DCF is estimated to be about 940 tons [19]. DCF has been detected in the groundwater and surface water in different regions such as Germany, Pakistan, Spain, Europe, China, and other regions [20]. The removal rate of DCF in wastewater treatment is only 21% to 40%, which poses serious threats to human health and aquatic ecology, such as gastrointestinal damage to humans, and hepatotoxicity and reproductive defects to marine animals [21,22]. DCF has been included in the surface water observation list of EU Resolution 2015/495, as it may cause more serious consequences when combined with other pollutants [23].

In this study, DCF was used as a model pollutant, and PS and PBAT were used as model MPs to compare the adsorption properties of different MPs on DCF before and after aging. The possible adsorption mechanism was proposed by combining the effect results of kinetics, isotherms, and environmental factors on the adsorption of DCF by MPs. The results show that surface aging can increase the adsorption capacity of MPs to organic pollutants. The adsorption process involves hydrophobic, electrostatic, and hydrogen bonding forces and other forces, which are also the main reasons for the adsorption differences before and after aging. These findings have important implications for the environmental fate of pollutants in the presence of MPs and the potential danger of MPs as carriers of micropollutants.

## 2. Materials and Methods

### 2.1. Materials

PS and PBAT with particle sizes ranging within 75–150 μm were selected for the experiments and purchased from China Hengfa Plastic Technology Co. Ltd. The chemical structures and properties of PS and PBAT can be found in Appendix A. The MPs were ultrasonically cleaned with deionized water for 5 min each time and repeated 3 times. The cleaned MPs were placed in a drying oven at 50 °C for 12 h and then stored. DCF was purchased from China Aladdin Industries, Inc, and its structure and properties are shown in Appendix A. Humic acid (HA) was supplied by China Beijing Dometic Technology Co. Sodium hydroxide (NaOH), hydrochloric acid (HCl), and sodium chloride (NaCl) were purchased from China Aladdin Industries. Deionized water was used in all experiments, and the purity of the reagents not mentioned was in analytical purity. The appropriate concentrations of the solutions could be obtained by following these steps: accurately weigh 500.0 mg of DCF reagent into methanol, dissolve it completely, transfer it to a 25 mL volumetric flask for volume fixation to obtain a concentration of 20 g/L of DCF stock solution, and then place it under shade and set aside; weigh 4.0 g of NaOH and dissolve it in distilled water, then transfer it to a 1000 mL volumetric flask and fix the volume to 1000.0 mL to get 0.1 mol/L NaOH solution; and lastly, transfer 8.3 mL of 12 mol/L concentrated hydrochloric acid to distilled water, then transfer it to a 1000 mL volumetric flask and fix the volume to 1000.0 mL to get 0.1 mol/L HCl solution.

### 2.2. Preparation and Aging of MPs

The aging of the MPs was performed in the UV aging chamber, and the aging process was performed using a high temperature and UV aging simultaneously. During the irradiation process, the MPs were mixed every 6 h to ensure uniform exposure. The temperature was set to 70 °C and the wavelength to 254 nm. The UV intensity was 30 mW/cm^2^ and the distance was 14 cm. The pristine PS and PBAT plastics were evenly laid flat in the Petri dishes and the aging lasted for 15 d. After aging, the MPs were washed three times with deionized water, dried at room temperature, and reserved for use.

### 2.3. Characterization of MPs

To characterize the MPs, the samples were vacuum-dried for at least 3 days before use. The surface morphology of MPs was characterized using scanning electron microscopy (SEM) from Japan. Fourier transform infrared (FTIR) from Germany spectroscopy was used to study the changes in surface structure and the functional groups of MPs by aging and by adsorption in the range of 400–4000 cm^−1^. The oxygen content of the MPs’ particles by UV aging was studied using an energy dispersive spectrometer (EDS) from Japan. The contact angle tested the change in hydrophobicity before and after aging. The differences in crystallinity between MPs were examined by X-ray diffraction (XRD, XRD-7000 s/L) from Germany with a scan range 2θ of 10~70°. The zeta potential can determine the trend of the surface potential of MPs with pH.

### 2.4. Adsorption Experiments

The intermittent equilibrium method was used to evaluate the adsorption capacity of MPs on DCF. All adsorption experiments were performed using 60 mL brown glass bottles with 50 mL solution volume, pH adjusted to 7.0, and shaken at 25 ± 1 °C and 160 rpm. The methanol concentration was kept below 0.1% (*v*/*v*) to avoid co-solvent interactions. The 20 g/L DCF stock solution was pipetted (50 μL) and fixed to 50 mL so that the DCF concentration was 20 ppm. The mass of the MPs was accurately weighed to 10.0 mg. In the process of the kinetic experiments, nine time points (0, 0.5, 1, 3, 6, 9, 12, 24, 36 h) were selected as the detection points. The adsorption equilibrium time of DCF on MPs is about 12 h, indicating that 24 h is enough to reach the equilibrium. Therefore, 24 h was selected as the reaction time in the subsequent experiments. In the adsorption isotherm experiments, different initial concentrations of DCF solution were prepared so that the initial concentration of DCF was 15–35 mg/L and 10 mg of PS as well as PBAT were added to 50 mL of DCF solution of different concentrations.

The effects of HA, pH, and salinity on the adsorption capacity of MPs were investigated in the experiment. The pH was adjusted with 0.1 mol/L of HCl and NaOH to control the pH from 3 to 9. The adsorption capacity of DCF on MPs was measured in different pH solutions, different concentrations of HA solution (0–20 mg/L), and different salinity (NaCl concentration of 0–0.6 mol/L). At the same time, blank controls without plastic and with plastic but without contaminants were also measured. After standing for half an hour, the supernatant was collected using a 0.22 μm filter and analyzed on a UV-visible spectrophotometer. The detection wavelength of DCF that was detected on the UV spectrophotometer was 275 nm; and the blank control of spectrophotometry was 0.1% methanol blank solution.

## 3. Results and Discussion

### 3.1. Characteristic Analysis of Aged MPs

#### 3.1.1. SEM Analysis

The MPs showed obvious color change after UV aging. As shown in Appendix A, yellowing of MPs occurs and the color change becomes more and more obvious with the time increased. The SEM images as shown in Figure 1 indicate that PS and PBAT have different profiles and surface morphologies before and after aging. It can be clearly observed that the surface of the microplastic becomes rough from smooth, and folds and cracks are produced. In addition, the erosion degree and the surface roughness of PBAT after aging are significantly stronger than that of PS.

#### 3.1.2. FTIR and EDS Analysis

The changes of the functional groups of PS and PBAT before and after aging were analyzed by FTIR, and the results are shown in Appendix A. It can be seen from the spectrum that PBAT shows strong peak intensities at 720 (C-H_3_), 1050 (O-C-O), 1370 (C-OH), 1720 (C=O), and 2960 (C-H) cm^−1^. PS shows strong peak intensities at 750 (C-H_3_), 1045 (O-C-O), 1490 (C-H), 1600 (C-H), 2916 (C-H), and 3059 (C-H_2_) cm^−1^. Comparing the local magnification of PBAT and PS before and after UV aging, it can be seen from Figure 2 that PBAT has peak intensity changes at 1370, 1602, and 1718 cm^−1^ after aging, corresponding to C-OH, C=C, and -COOH, respectively. Meanwhile, aged PS mainly shows peak changes at 1666 cm^−1^, corresponding to C=O. From the above results, it can be concluded that the oxygen-containing functional groups of PBAT and PS increased after UV irradiation. EDS results as shown in Figure 3 show that the oxygen content of PBAT increases from 35.68% to 44.37%, and PS increases from 8.51% to 23.72%. The data listed in Appendix A further demonstrates that the oxygen content of MPs increased after UV aging, and the increase of PS was more obvious than that of PBAT, which is consistent with the above FTIR results.

#### 3.1.3. Zeta Potential and XRD Analysis

Figure 4 shows the variation trend of MPs’ surface potential with pH. The XRD spectra can well-reflect the crystallization degree of MPs. Polymers with high crystallinity usually have sharp diffraction peaks. As shown in Figure 5, three sharp diffraction peaks appear in the XRD pattern of PBAT, indicating the high crystallinity of PBAT. Similarly, a sharp diffraction peak appears in the XRD spectrum of PS. The peak intensity of high and low judges its crystallinity level. According to the XRD patterns of PBAT and PS, the crystallinity of the MPs before aging is higher than that after aging. By comparing the two figures, a and b, it can be seen that the order of crystallinity is PS > PBAT > A-PS > A-PBAT. Therefore, the crystallinity of the aged MPs particles decreases. Figure 6 shows the contact angles of PS and PBAT before and after aging. It can be seen that the contact angles of PS range from 143° to 84.49°, and those of PBAT range from 87.86° to 41.01°. The pristine MPs’ surface is non-wetting and highly hydrophobic, but the MPs become more hydrophilic after aging because new hydrophilic groups may be generated on the MPs’ surface, which is consistent with the FTIR analysis. In summary, the aging process of PS and PBAT has been speculated, as shown in Appendix A [24,25].

### 3.2. Adsorption Kinetics

Figure 7a,b show the adsorption kinetics of PBAT and PS for DCF. The kinetic models are found in the Appendix A. At the beginning of the experiment, the adsorption of DCF by MPs increased rapidly and reached equilibrium at 12 h. The equilibrium adsorption amount of DCF on MPs are: Q_(A-PBAT)_ (27.65 mg/g) > Q _(A-PS)_ (23.91 mg/g) > Q _(PBAT)_ (9.30 mg/g) > Q _(PS)_ (9.21 mg/g). Therefore, in the subsequent experiments, the reaction time was set as 24 h. The behavior of DCF adsorption by MPs was fitted using a pseudo-first-order kinetics model (dotted line) and pseudosecondary kinetics model (solid line), and the fitting results are shown in Appendix A. The first-order kinetics model is widely used for adsorption processes that are dominated by physical diffusion control. The pseudosecondary adsorption model indicates that the adsorption process is dominated by chemisorption [26]. This suggests that the adsorption of DCF by MPs is dominated by chemisorption.

### 3.3. Adsorption Isotherm

The adsorption isotherm model can predict the interaction between pollutants and adsorbents at the adsorption equilibrium. The adsorption isotherm models are found in the Appendix A. Fitting isothermal sorption data was done using the Langmuir and Freundlich models (Figure 7c,d). All fitted plots and isotherm parameters for DCF adsorption on MPs are shown in Figure 6 and Appendix A, respectively. From Appendix A, it can be concluded that the R^2^ values of the Freundlich adsorption model for DCF adsorption by MPs are all greater than those of the Langmuir adsorption model. Meanwhile, the adsorption amount for DCF was Q _(A-PBAT)_ > Q _(A-PS)_ > Q _(PBAT)_ > Q _(PS)_, which is consistent with the experimental results. In summary, the adsorption isotherms are consistent with the Freundlich model. The Freundlich isotherm model is an empirical model in which adsorption is non-uniform, while Langmuir isotherm adsorption is the uniform adsorption of a single molecular layer with the same affinity for the adsorbent [27]. Therefore, non-uniform adsorption plays a role in the process of DCF adsorption by MPs. Du et al. showed that the adsorption of RhB on MPs is more suitable for the Langmuir adsorption model [28], while Shi et al. found that the Freundlich model in the adsorption process was more suitable than the Langmuir model [18]. These studies indicate that the adsorption properties of MPs are related to the type and nature of organic compounds.

### 3.4. Effects of Environmental Factors on Adsorption

#### 3.4.1. Effect of pH

The change of solution pH will influence the surface charge of MPs and the existence state of diclofenac sodium in solution, thus affecting the adsorption of DCF by the MPs [29]. As shown in Figure 8a, the adsorption of MPs on DCF decreased with increasing pH and decreased rapidly at 4 to 5. The results show that MPs have a high adsorption capacity for DCF at low pH conditions, and the adsorption capacity decreases gradually with the pH increased. Sun et al. also verified that acidic solutions facilitate the adsorption of contaminants on MPs [30].

The effect of pH on the adsorption of contaminants by MPs can be explained by electrostatic interaction. When the pH is lower than the p*K*a of DCF (4.15), DCF exists in aqueous solution in the nonionized form. When the pH is higher than the p*K*a, the proportion of the nonionized form of ionized DCF decreases with increasing pH while the proportion of the anionic form increases with increasing pH [31]. Moreover, from Figure 8a, the adsorption of DCF by MPs decreased abruptly when the pH changed from 4 to 5. This may be due to the fact that the DCF changed from the molecular form to the anionic form at this time, and the negative charge on the MPs’ surface leads to a sudden increase in electrostatic pulse between anionic DCF and MPs. Therefore, when the pH value of the solution is higher than 4, the adsorption capacity of MPs to DCF decreases. DCF has a log Kow value of 4.51, which is a hydrophobic pollutant, and PS as well as PBAT are hydrophobic substances [25]; therefore, the hydrophobic effect is the main role of the adsorption process. After aging, the hydrophobicity of PS and PBAT decreases, but the adsorption capacity of DCF increases, indicating that hydrophobicity plays an important but not decisive role in the adsorption process. Wang et al. reported that the adsorption of hydrophobic organic compounds by MPs is mainly through hydrophobic and π-π interactions [32], and DCF was adsorbed on the adsorbent through hydrophobic, electrostatic, and π-π interactions [33,34,35]. Thus, the high adsorption mechanisms of MPs on neutral DCF at a pH lower than 4 are mainly due to hydrogen bonding and hydrophobic interactions. Meanwhile, electrostatic interactions have a greater influence when pH is higher than 4, which is also consistent with the experimental results.

#### 3.4.2. Effect of HA

Figure 8b shows the adsorption properties of MPs for DCF with different concentrations of HA. The addition of HA reduced the adsorption of DCF on MPs and the higher the concentration of HA, the lower the adsorption capacity. HA molecules carry their own functional groups that can interact with MPs or pollutants, thus influencing the adsorption performance of MPs. In addition, HA molecules are organic macromolecules, which may occupy the adsorption sites on the surface of MPs before DCF. The adsorption capacity of MPs to DCF in pure water is 8.735 mg/g (PS), 9.406 mg/g (PBAT), 23.068 mg/g (A-PS), and 26.428 mg/g (A-PBAT), while in 35 mg/L HA solution, the capacity decreases to 2.800, 4.367, 17.469, and 21.053 mg/g, respectively. Compared with pure water, the adsorption capacity of PS decreases the most (67.95%) before aging, and the adsorption capacity of PBAT decreases the least (20.34%) after aging. It is possible that HA interacts with DCF through a complexation hydrophobic interaction, leading to the decrease in adsorption capacity of DCF on MPs. The adsorption properties of PS and PBAT before and after aging were inhibited with the addition of HA, and the inhibition of PBAT after aging is relatively small, but the inhibition of original PS is the largest. It can be seen from the SEM plots that compared with other MPs, the aged PBAT has more folds and pores, a larger specific surface area, as well as more adsorption sites with less inhibition. Meanwhile, the original PS has fewer adsorption sites, which preferentially occupies the adsorption sites when the large molecules enter, making the adsorption amount much lower. Therefore, the active sites on the surface of MPs play an important role in the adsorption of DCF by MPs.

#### 3.4.3. Effect of Ionic Strength

NaCl is widely present in various waters and has an important effect on the adsorption of organic pollutants on MPs [36]. The average salinity of seawater is 35‰ (0.62 M). Therefore, salinities of 0–0.60 M (0, 0.15, 0.30, 0.45, 0.60 M) were selected to simulate seawater environment in this experiment to study the adsorption process and the difference of pollutants by MPs in fresh water and in seawater environment, and the results are shown in Figure 7c,d. As the salinity increased, the Na^+^ ions in solution competed with DCF for the adsorption sites on the surface of MPs, resulting in a gradual decrease in the adsorption of MPs on DCF [37]. Meanwhile, the presence of Na+ neutralized the negative charges on the adsorption sites of MPs, which weakened the electrostatic interactions between MPs and DCF [38]. The adsorption capacity of DCF by MPs in pure water was 11.602 mg/g (PS), 12.484 mg/g (PBAT), 16.015 mg/g (A-PS), and 23.834 mg/g (A-PBAT), while in 0.6 M NaCl solution, the capacity decreased to 3.783, 5.296, 8.197, and 4.035 mg/g, respectively. The highest decrease compared to pure water was 83.07% for PBAT after aging and the lowest was 34.3% for PBAT after aging. Chen and co-workers also confirmed that salinity can inhibit the adsorption of contaminants by MPs, which is consistent with the conclusion of this study [39]. In summary, MPs in freshwater environments may be more hazardous and have a higher sorption capacity for certain organic pollutants.

### 3.5. Adsorption Mechanism

Properties such as hydrophobicity and crystallinity of MPs play an important role in the adsorption mechanism [40]. DCF is a hydrophobic organic matter, and the original MPs are hydrophobic and have a certain adsorption effect on DCF. After aging, the MPs become hydrophilic, but the adsorption of DCF increases significantly, indicating that the hydrophobic effect occupies a certain position in the adsorption of DCF by MPs, but it is not decisive. Since the hydrophilic groups of aged MPs increased with the increase of diclofenac sodium adsorption, it is hypothesized that hydrogen bonding, halohydrogen bonding, and π-π interactions have a significant influence in the adsorption process. For pristine and aged PS and PBAT, the broad characteristic peaks in the range of 3500–3100 cm^−1^ are associated with the vibration of hydroxyl or carboxyl groups and intermolecular hydrogen bonding [41]. The results indicate that intermolecular hydrogen bonding is a possible adsorption mechanism. The fitted data for isothermal adsorption according to the Langmuir and Freundlich models are more consistent with the Freundlich model, indicating that the adsorption of DCF by MPs is non-uniform adsorption.

In addition, the adsorption of DCF by aged MPs is related to the pH and ionic strength of the solution. Based on the above discussion, a reasonable adsorption mechanism of pristine and aged PS and PBAT on DCF was proposed, as shown in Figure 9. At low pH (pH < pH_PZC_), the surfaces of PS and PBAT MPs are positively charged, while DCF is in the molecular form and uncharged at this time. The main adsorption process is controlled by the hydrogen-halogen bonds formed by the interaction between nitrogen, oxygen, and chlorine atoms on DCF, and hydrogen atoms on MPs. On the other hand, at higher pH (pH > pH_PZC_), the surface charge of MPs is negative, and DCF exists in the anionic state. At this time, the adsorption of MPs to DCF decreases due to electrostatic interaction. However, as can be seen from Figure 7a, when the pH value is low, the adsorption capacity is much larger than that when the pH value is high, and it drops sharply between pH 4 and 5. Therefore, the electrostatic effect is a relatively important factor in the adsorption of DCF by MPs, PS and PBAT, and hydrogen bonding also plays a role in the adsorption process. Therefore, for anionic DCF, with the increase in salinity, the adsorption capacity decreases with the exchange of cations. When the pH is higher than 4, the MPs are negatively charged and Na^+^ is positively charged to replace the diclofenac ions adsorbed to the MPs, which further confirms the electrostatic interaction between DCF and MPs. In summary, electrostatic interactions as well as hydrogen-halogen bonding play major roles in the adsorption process.

## 4. Conclusions

The adsorption of DCF on MPs before and after aging and its influencing factors were investigated. The adsorption order of PS and PBAT to DCF before and after aging is Q_(A-PBAT)_ (27.65 mg/g) > Q _(A-PS)_ (23.91 mg/g) > Q _(PBAT)_ (9.30 mg/g) > Q _(PS)_ (9.21 mg/g). The adsorption capacity of aged MPs on DCF increased, and the adsorption equilibrium time of PS and PBAT before and after aging was about 12 h. The pseudosecondary kinetic model could better describe the adsorption kinetics compared with the pseudo-first-order kinetic model. The adsorption isotherms of different types of MPs and MPs before and after aging are different, and the adsorption model of MPs on DCF was more suitable for the Freundlich adsorption model than the Langmuir adsorption model. The adsorption process of MPs on DCF is highly dependent on the pH, ionic strength, and humic acid strength of the solution. The adsorption performance of MPs on DCF was better under acidic conditions, and both salinity and humic acid strength could inhibit the adsorption of DCF on MPs. In conclusion, during the adsorption of DCF by MPs, the adsorption performance decreases with the increase in pH, and the increase in salinity and humic acid concentration can inhibit the adsorption of DCF by MPs. The results show that the adsorption process is mainly controlled by hydrophobic, electrostatic, and π-π interactions, in which electrostatic and hydrogen-halogen bonds occupy the main position.

## Figures and Tables

**Figure 1 toxics-11-00024-f001:**
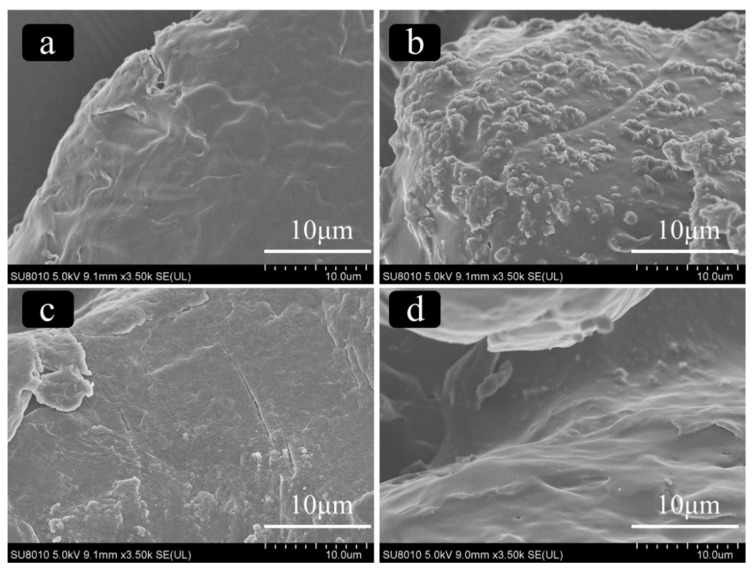
SEM images of the surface microstructure of PS and PBAT before and after UV aging ((**a**) pristine PBAT, (**b**) aged PBAT, (**c**) pristine PS, and (**d**) aged PS).

**Figure 2 toxics-11-00024-f002:**
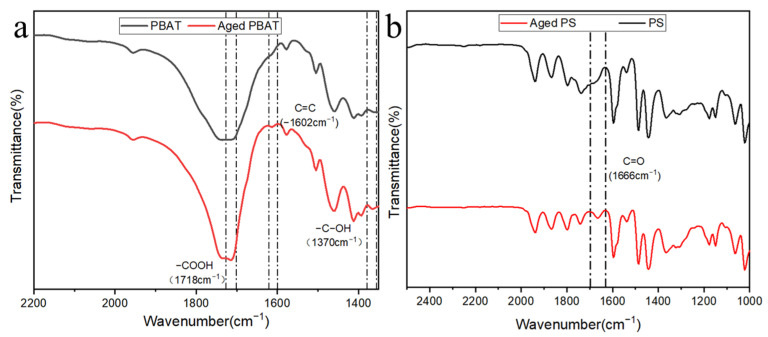
FTIR spectra before and after UV aging and adsorption ((**a**) PBAT and (**b**) PS).

**Figure 3 toxics-11-00024-f003:**
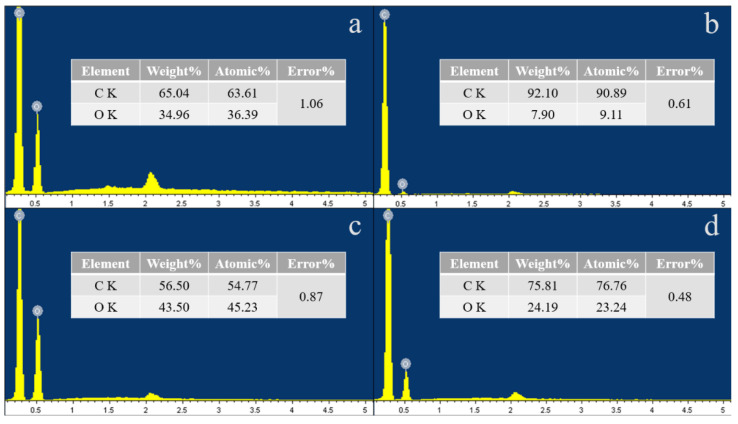
Changes in the oxygen content of PBAT and PS MP particles before and after UV aging ((**a**) P-PBAT, (**b**) P-PS, (**c**) A-PBAT, and (**d**) A-PS).

**Figure 4 toxics-11-00024-f004:**
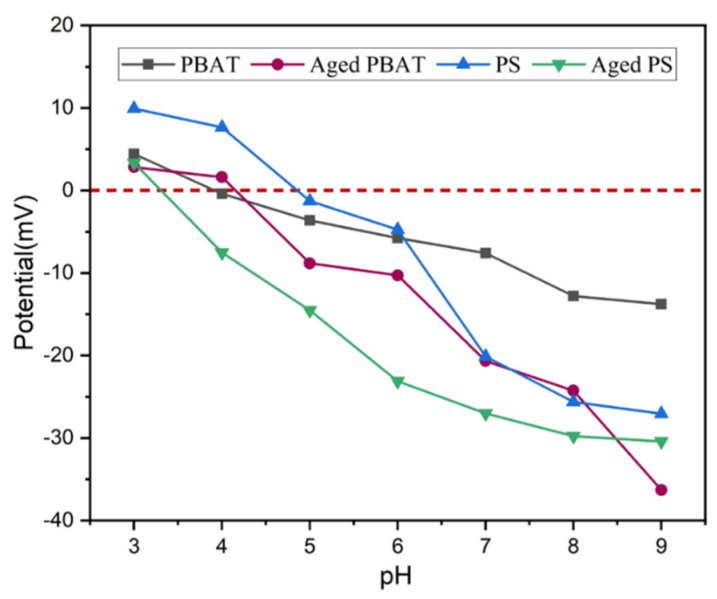
Zeta potential of MPs.

**Figure 5 toxics-11-00024-f005:**
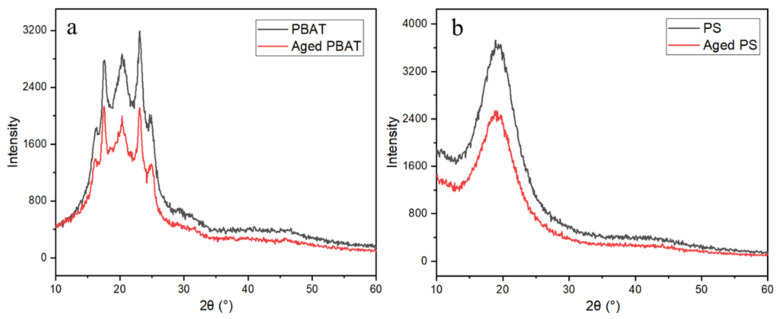
XRD spectra of PBAT and PS ((**a**) PBAT, (**b**) PS).

**Figure 6 toxics-11-00024-f006:**
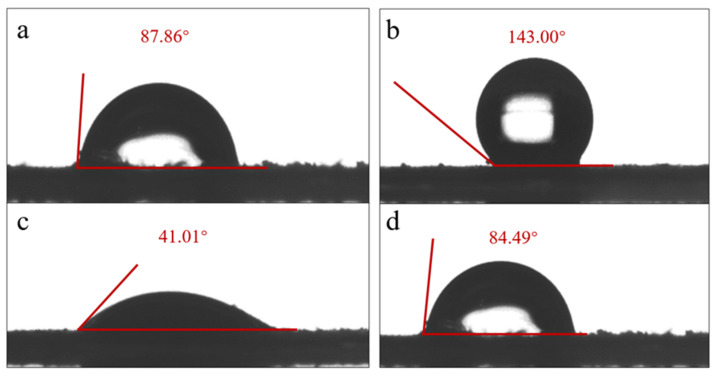
Contact angle results of MPs before and after UV aging ((**a**) P-PBAT, (**b**) P-PS, (**c**) A-PBAT, and (**d**) A-PS).

**Figure 7 toxics-11-00024-f007:**
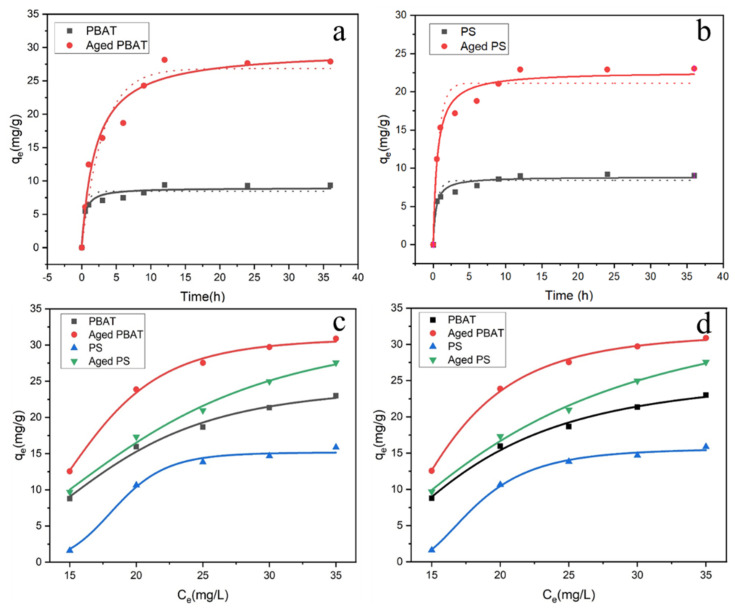
Adsorption kinetics of DCF by MPs ((**a**) PBAT, (**b**) PS), and the adsorption isotherms ((**c**) Langmuir model, (**d**) Freundlich model).

**Figure 8 toxics-11-00024-f008:**
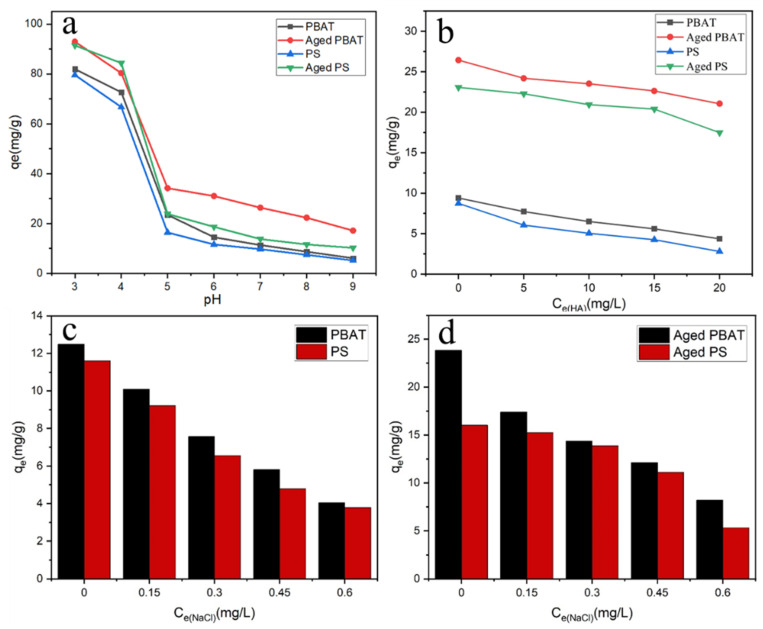
Effect of environmental factors on DCF adsorption by MPs ((**a**) pH, (**b**) HA, (**c**) pristine MPs- salinity (**d**) aged MPs- salinity).

**Figure 9 toxics-11-00024-f009:**
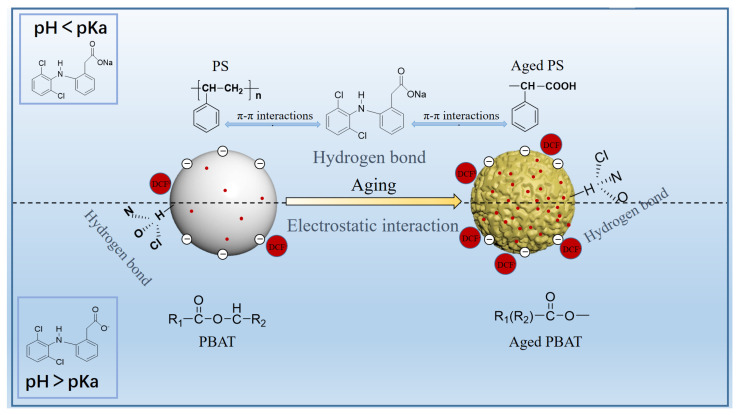
Mechanism of DCF adsorption by MPs.

## Data Availability

Not applicable.

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
