# Peer review of "Adsorption of Diclofenac Sodium by Aged Degradable and Non-Degradable Microplastics: Environmental Effects, Adsorption Mechanisms"

_toxics, 2022, doi:10.3390/toxics11010024_

Round 1
Reviewer 1 Report
I have read the manuscript entitled ‘Effect of aging on adsorption behavior of polystyrene and 2 poly(butylene adipate-co-terephthalate) microplastics for diclo- 3 fenac sodium: Adsorption mechanism and aging’ My comments and suggestions are mentioned below. Please consider carefully during the revision process of your manuscript.
1. Please concentrate/rewrite the title of the manuscript, to be more attractive for readers.
2. Follow the writing instructions of the journal (affiliation, figures, references, etc.).
3. Rewrite the following sentences, The equilibrium time for the adsorption of DCF by MPs was about 12 h, and 24 h was sufficient to reach the adsorption equilibrium. In the subsequent experiments, the samples were shaken for 24 h in a thermostatic shaker to 138 ensure the equilibrium.”
4. Divide 3.1. Characteristic analysis of aged MPs in: 3.1.1 SEM analysis; 3.1.2. FTIR analysis, …. XRD analysis, etc., Also, for each parameter studied in Results and Dissections please add Figures to be easy understand by the readers.
5. In Conclusion section, write a conclusion for each parameters studied.

Author Response
1. Please concentrate/rewrite the title of the manuscript, to be more attractive for readers.
Reply: Thank you for your advice. We changed the title into:“Adsorption of diclofenac sodium by aged degradable and non-degradable microplastics: Environmental effects, adsorption mechanisms”。
2. Follow the writing instructions of the journal (affiliation, figures, references, etc.).
Reply: Thanks for your reminding. We have made the changes as requested.
- Rewrite the following sentences, the equilibrium time for the adsorption of DCF by MPs was about 12 h, and 24 h was sufficient to reach the adsorption equilibrium. In the subsequent experiments, the samples were shaken for 24 h in a thermostatic shaker to 138 ensure the equilibrium.”
Response: Thank you for your careful inspection. The sentence has been modified as: The adsorption equilibrium time of DCF on MPs is about 12 h, indicating that 24 h is enough to reach the equilibrium.
- Divide 3.1. Characteristic analysis of aged MPs in: 3.1.1 SEM analysis; 3.1.2.
FTIR analysis, …. XRD analysis, etc., Also, for each parameter studied in Results and Dissections please add Figures to be easy understand by the readers.
Reply:Thank you for your advice. We modified the relevant content and matched the corresponding data graph.
- In Conclusion section, write a conclusion for each parameters studied.
Reply:Based on your comments, a conclusion has been added for each parameter of the study in the conclusion section.

Reviewer 2 Report
The manuscript entitled ‘Effect of aging on adsorption behavior of polystyrene and poly(butylene adipate-co-terephthalate) microplastics for diclofenac sodium: Adsorption mechanism and aging’ addresses the very important and up-to-date issue of the existence of both microplastics and personal care products in the environment. However, I do have several suggestions/comments, which might be further helpful in improving the content of the current article.
1. Authors should subject MPs to nitrogen adsorption/desorption studies to estimate specific surface area and pore distribution.
2. Line 150-151: please specify what was the blank during the spectrophotometric measurement.
3. Line 178-180: How was the order of crystallinity determined? The order given seems strongly approximate and unsupported by any calculations.
4. Fig. 3: Kindly change the background of the figure to white.
5. Line 328: ‘the surfaces of PS and PBAT MPs are positively charged’ - While in Figure 9 there is a negative charge on the model MPs surface.
6. Fig. 9: please explain what the red balls mean. Moreover, please mark in the figure π-π interactions.
7. Did the authors check the possibility of regeneration of MPs after the sorption process?
8. Table S2: ‘Structure and properties of the diclofenac sodium’ - the formula given in the table is not the formula of diclofenac sodium.
9. Fig. S3: Please explain what is R1 and R2 in the second stage, and R1 in the third stage.
Round 2
Reviewer 1 Report
Accept in present form.